**Data Availability Statement:** All relevant data are within the paper and its Supporting Information files.

# Trends in perioperative practices of high-risk surgical patients over a 10-year interval

**Brenno Cardoso Gomes**[1,2‡]*, **Suzana Margareth Ajeje Lobo**[3☯], **Luiz Marcelo Sá Malbouisson**[1☯], **Renato Carneiro de Freitas Chaves**[4☯], **Thiago Domingos Corrêa**[4☯], **Cristina Prata Amendola**[5☯], **João Manoel Silva Júnior**[1,4‡], on behalf of The BraSIS research group[¶]

**1** Hospital das Clínicas, Faculdade de Medicina da Universidade de São Paulo (USP), São Paulo-SP, Brasil, **2** Departamento de Medicina Integrada do Setor de Ciências da Saúde da Universidade Federal do Paraná, Curitiba-PR, Brasil, **3** Hospital de Base de São José do Rio Preto (FAMERP), Faculdade de Medicina, São José do Rio Preto-SP, Brasil, **4** Hospital Israelita Albert Einstein, São Paulo-SP, Brasil, **5** Hospital de Câncer de Barretos, Fundação Pio XII, Barretos-SP, Brasil

☯ These authors contributed equally to this work.
‡ These authors also contributed equally to this work
¶ Membership of the BraSIS research group is provided in the Acknowledgments
* brennogomes@ufpr.br

## Abstract

### Introduction

In Brazil, data show an important decrease in morbi-mortality of high-risk surgical patients over a 10-year high. The objective of this post-hoc study was to evaluate the mechanism explaining this trend in high-risk surgical patients admitted to Brazilian ICUs in two large Brazilian multicenter cohort studies performed 10 years apart.

### Methods

The patients included in the 2 cohorts studies published in 2008 and 2018 were compared after a (1:1) propensity score matching. Patients included were adults who underwent surgeries and admitted to the ICU afterwards.

### Results

After matching, 704 patients were analyzed. Compared to the 2018 cohort, 2008 cohort had more postoperative infections (OR 13.4; 95%CI 6.1–29.3) and cardiovascular complications (OR 1.5; 95%CI 1.0–2.2), as well as a lower survival ICU stay (HR = 2.39, 95% CI: 1.36–4.20) and hospital stay (HR = 1.64, 95% CI: 1.03–2.62). In addition, by verifying factors strongly associated with hospital mortality, it was found that the risk of death correlated with higher intraoperative fluid balance (OR = 1.03, 95% CI 1.01–1.06), higher creatinine (OR = 1.31, 95% CI 1.1–1.56), and intraoperative blood transfusion (OR = 2.32, 95% CI 1.35–4.0). By increasing the mean arterial pressure, according to the limits of sample values from 43 mmHg to 118 mmHg, the risk of death decreased (OR = 0.97, 95% CI 0.95–0.98). The 2008 cohort had higher fluid balance, postoperative creatinine, and volume of intraoperative

**Funding:** The author(s) received no specific funding for this work.

**Competing interests:** The authors have declared that no competing interests exist.

blood transfused and lower mean blood pressure at ICU admission and temperature at the end of surgery.

## Conclusion

In this sample of ICUs in Brazil, high-risk surgical patients still have a high rate of complications, but with improvement over a period of 10 years. There were changes in the management of these patients over time.

## Background

High-risk surgical patients represent a small and specific group, ranging from to 10% to 12,5% of the overall surgical population, with a high morbimortality [1]. In a report from Pearse et al., the mortality rate in general surgical population was 1,9% in the United Kingdom, while in the high-risk patients was 83.8% [1]. In this high-risk population, postoperative complications lead to death in shorter time courses than in the general surgical patients [2].

Advances in perioperative care practices supported by evidence-based medicine during the last decade were associated with a reduction in postoperative morbimortality worldwide [3]. In Brazil, two studies evaluated the outcomes of high-risk surgical patients with an interval of 10 years [4,5]. It was observed a reduction in surgical mortality in high-risk patients from 20% to 9.6% and in complications from 38% to 29.9%. The first cohort evaluated outcomes of non-cardiac surgical patients in 21 Brazilian ICUs (intensive care units), from 18 institutions (eight public hospitals and 10 private hospitals) was collected in 2006 [5]. The second cohort evaluated outcomes of high-risk surgical patients admitted to 29 ICUs during 2017 [4].

In order to understand the mechanisms and determinants associated to the improvement in the high-risk surgical patient's outcomes during the last decade, we performed a post-hoc comparison of the databases of two large multicenter Brazilian studies. Therefore, the objective of this study was to compare the prognostic trends of high-risk surgical patients admitted to Brazilian ICUs, through the analysis of two large Brazilian multicenter studies separated by an interval of 10 years.

## Methods

### Study design and participating centers

This post-hoc analysis compared two prospective multicenter cohorts on the epidemiology and clinical outcomes of high-risk surgical patients in Brazil. The first cohort study was conducted from 2006 to 2008 [5], and the second study was conducted from 2017 to 2018 [4]. To facilitate the reading and understanding of the content, throughout most of this paper, we refer to the studies as the 2008 and 2018 cohorts, the years in which the works were finalized. Adult patients (aged ≥ 18 years) undergoing elective or emergency surgery were admitted to the ICU after surgery. This project was approved by the Research Ethics Committee of the School of Medicine—University of São Paulo (FMUSP) and registered in the National System of the Brazil Platform (CAAE)—CAAE: 30690820.1.0000.0068. The research ethics committee waived the application of the consent form, as this is a retrospective study with exclusive analysis of medical records with a database.

Because the criteria for ICU admission in the postoperative period were not standardized between the centers, all patients with indications for postoperative ICU admission were

considered high risk surgical patients. Each patient was admitted to hospitals with expertise in the care of that patient profile.

The exclusion criteria of the two studies were very similar and were adjusted in the composite database to maintain sample homogeneity. Therefore, patients undergoing cardiac surgery, obstetric patients, pregnant women, and palliative patients with no prospect of curing their disease were excluded from the two studies.

## Patients and variables

The following data were collected: demographic data (age, sex, ethnicity, and body mass index), beside preoperatively we composed classification of physical status (American Society of Anesthesiologists, ASA) and comorbidities (systemic arterial hypertension, heart disease, cancer, diabetes mellitus, smoking, chronic obstructive pulmonary disease, alcoholism, chronic renal failure, unspecified stroke). Furthermore, the worst values of mean arterial pressure (MAP), heart rate, axillary temperature, arterial lactate, hemoglobin, hematocrit, Sequential Organ Failure Assessment (SOFA) score at ICU admission and characteristics of surgeries (emergency or elective, topographic surgery, and time of surgery), occurrence of ICU complications, length of stay in the ICU, length of stay in the hospital, and mortality in the ICU, in the hospital, and within 30 days after discharge were analyzed in study.

All patients were followed up until hospital discharge and 30 days postoperatively, and the following complications were evaluated during their stay in the ICU. The presence of complications was described by the investigators of both studies and recorded in a database.

In addition, intraoperative fluid balance, the use of vasoactive drugs, and the presence of mechanical ventilation at ICU admission were analyzed. The fluid balance was calculated by quantifying the volume administered minus urine output and tube output.

**Definition of postoperative complications.** Cardiovascular: characterized by the need for vasopressors for more than 1 hour despite adequate volume resuscitation; acute myocardial infarction; arrhythmias; or cardiac arrest.

Respiratory: a relationship between partial pressure of oxygen and fraction of inspired oxygen (PaO2/FiO2) < 200 in patients without previous heart disease; the need for reintubation; or the presence of bronchospasm or pneumothorax.

Neurological: Richmond Agitation and Sedation Scale (RASS) score that acutely fluctuates and agitation as determined by RASS $\geq$ +2, documented convulsive seizures or stroke.

Coagulation: reduction of platelet counts greater than 30% of the baseline value during the preoperative period, platelet count below 100,000mm$^3$, or acute bleeding above 100 mL/hour associated with a decrease of 3 hematocrit points.

Gastrointestinal: presence of acute abdominal distension, uncontrolled nausea, and vomiting, need for parenteral nutrition, more than three episodes of diarrhea within 24 hours, acute gastrointestinal bleeding, acute liver failure, acute pancreatitis or presence of moderate- to high-output fistulas.

Infection: as the definitions of infection were different in 2008 cohort compared to 2018, we pragmatically chose to unite the classificatory terms for each period and sum the events. Thus, if there was any classifying type of infection in addition to the use of antibiotics and any positive cultures, it was considered a confirmed event, regardless of the severity of the event.

**Outcomes.** The main objective of this study was to analyze the mortality trends in high-risk surgical patients admitted to Brazilian ICUs in the period from 2008 to 2018. The secondary measures were differences in complication rates, the lengths of stays in the ICU and hospital, the need for mechanical ventilation in the immediate postoperative period, the use of

packed red blood cells, differences in the management of fluid replacement in these patients, use of vasoactive drugs in the immediate postoperative period and the hemodynamic parameters at ICU admission.

**Data storage.** The data were entered into an electronic database, Research Electronic Data Capture–REDCap [6,7], and then analyzed in the Statistical Package for the Social Sciences version 26.0 (SPSS Inc.®; Chicago, IL, USA) and version R v.3.4.1 (R Foundation for Statistical Computing, Vienna, Austria).

## Statistical analysis

Categorical variables are presented as absolute and relative frequencies. Quantitative variables were presented as mean and standard deviation or as the median and interquartile range, as appropriate. We used the Kolmogorov–Smirnov test to evaluate the distribution pattern of continuous numerical variables.

Proportions were compared using the chi-squared test, or Fisher's exact test if the assumptions for the use of chi-squared were violated. Quantitative variables were compared with the T-test or the Mann–Whitney test, as appropriate.

To compare data between the 2008 and 2018 cohorts, since the data of the groups were collected in two different periods in nonrandomized designs, differences in the baseline characteristics of the patients could lead to biased estimates of effects. Thus, to balance the baseline characteristics and reduce bias, we matched patients from the 2008 and 2018 cohorts using a propensity score, defined as the conditional probability of being treated. First, a logistic regression model was created using the group variable (2008 or 2018) as the dependent variable. Other potential confounding factors for morbidity and length of stay were considered in the analyses using propensity score matching (PSM) with a combination of variables: age, SAPS 3 (Simplified Acute Physiology Score 3), ASA, SOFA, time of surgery, type of surgery, comorbidities such as hypertension, diabetes mellitus, smoking, and previous anemia, which were included as predictors, and the tolerance width of the correspondence was established at 0.02 of the *logit*. These variables were selected in the pairing using a backdoor criterion, which detected the presence of confounding variables. The significant variables in the bivariate analyses and those considered clinically relevant were also subjected to the model. Next, a patient correspondence for each 2018 group was selected from the 2008 group based on the closest logit. This model was constructed based on a sample of patients matched by propensity score at a 1:1 ratio without replacement or repetition. The matching procedure was performed before analyzing the results of this study. The results of the model are expressed as odds ratios (ORs) and their 95% confidence intervals.

Survival analyses were performed using Cox proportional hazards models and Kaplan–Meier curves fitted to the Cox model. The survival time was computed in days. ICU and hospital survival estimates were calculated for the groups using the survival curves and were compared in the Cox risk model.

The association between explanatory variables and outcomes was evaluated with logistic regression models. The variables selected in the bivariate analyses ($P < 0.05$) and those considered clinically relevant were subjected to multiple logistic regression analysis. We evaluated the collinearity first by examining the dispersion matrix and the Pearson correlation coefficient for continuous variables or the cross-tabulation for categorical variables. We also ran a collinearity with analysis of the variance inflation factor. Variables with substantial collinearity were excluded. The results of the logistic regression analysis are expressed as ORs and 95% CIs. All the significance probabilities (P values) presented are two-tailed. P values $< 0.05$ were considered statistically significant.

The missing baseline data were treated with linear multiple imputation if it was less than 5%. The missing outcome variables were treated with pairwise deletion, a process in which information is eliminated when there is an absence of essential data for the test.

## Results

For this post-hoc analysis, 1905 patients were initially screened from the two cohort databases to be included. The 2008 cohort included 885 surgical patients admitted to 21 Brazilian ICUs and the 2018 cohort included 1020 patients in 29 ICUs, from all regions of the country. Two hundred ninety-eight patients of the 2008 cohort and 116 of the 2018 cohort were excluded from the analysis. The final composed cohort included 1491 patients. After PSM, 704 patients were analyzed (352 patients in each period, 2008 and 2018) (Fig 1).

Comparing the ICUs in each cohort, 55.5% were in private hospitals in 2008 while in 2018, 48.3% were in private institutions, making for 44.5% and 51.7% in public hospitals, respectively (P = 0.62). The differences in the regional distribution of the ICUs between the two periods were 4.0% in the North region (P = 0.36), 4.0% in the Northeast region (P = 0.68), 0.0% in the Midwest region (P = 1.00), 6.0% in the Southeast region (P = 0.67), and 2.0% in the South region (P = 0.85) (Fig 2).

In the combined sample, the age was 61.0 ± 17.4 years, and 51.4% were male. The 2018 cohort had higher SAPS 3 scores values and ASA physical status 3–4, while the 2008 study had higher SOFA scores values, ASA physical status score 2, chronic diseases, especially systemic arterial hypertension, cancer, chronic obstructive pulmonary disease and smoking. After PSM, 352 patients remained in each group, and all their baseline characteristics were well balanced (Table 1).

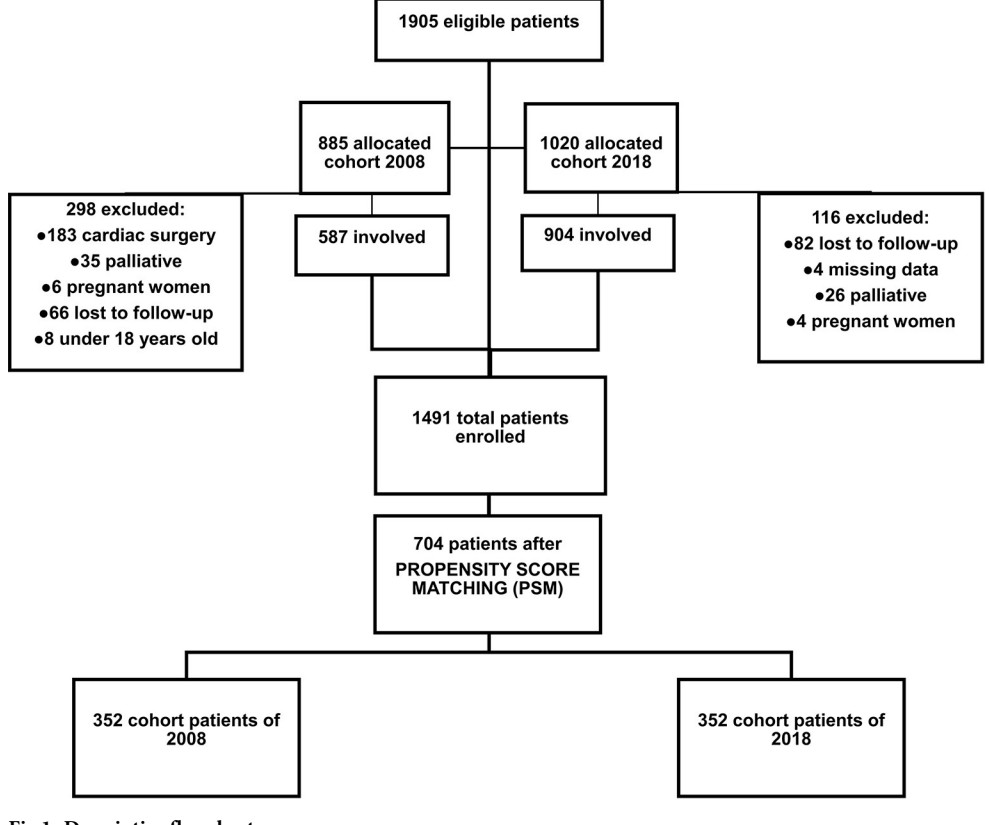

**Fig 1. Descriptive flowchart.**

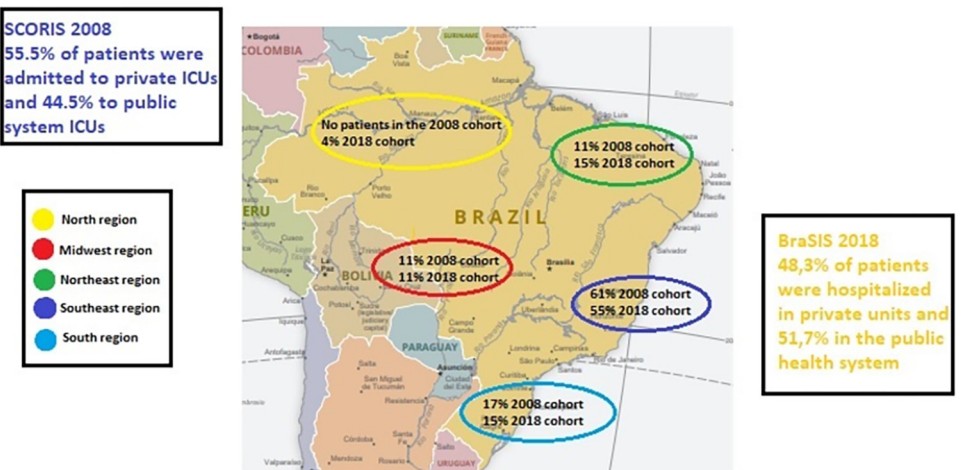

**Fig 2. Recruitment of patients from both cohorts per Brazilian region.** Political South America. Modified from CIA Factbook: https://www.cia.gov/the-world-factbook/static/57275a9c02995bfd31c79e1fa4887f27/south_america_pol.pdf.

**Table 1. Baseline characteristics of patients, with and without propensity score matching.**

| | Full Cohort | | | Matched Cohort | | | |
|---|---|---|---|---|---|---|---|
| | 2018 (n = 904) | 2008 (n = 587) | P | 2018 (n = 352) | 2008 (n = 352) | P | |
| Male, n (%) | 444 (53.8%) | 322 (54.9%) | 0.70 | 188 (57.3%) | 181 (51.4%) | 0.12 | |
| Age (year), mean± SD | 60.1± 17.7 | 62.4± 16.8 | 0.90 | 61.5±17.3 | 61.7±17.2 | 0.90 | |
| SAPS 3 score, mean± SD | 45.6±15.4 | 42.4±10.9 | < 0.01 | 42.1±15.1 | 41.6 ± 10.9 | 0.63 | |
| SOFA score (ICU admission), median (IQR) | 2.0 (1.0–5.0) | 4.0 (2.0–7.0) | < 0.01 | 3.0 (1.0–5.0) | 4.0 (2.0–6.0) | 0.09 | |
| ASA physical status score, n (%) | | | < 0.01 | | | 0.09 | |
| 2 | 405 (47.8%) | 285 (48.4%) | | 192 (54.5%) | 163 (46.3%) | | |
| 3 | 296 (34.9%) | 172 (29.3%) | | 103 (29.3%) | 117 (33.2%) | | |
| 4 | 61 (7.2%) | 39 (6.6%) | | 12 (3.4%) | 20 (5.7%) | | |
| Chronic diseases, n (%) | 707 (80.3%) | 479 (81.6%) | 0.55 | 303 (86.1%) | 289 (82.1%) | 0.15 | |
| Hypertension | 396 (44.2%) | 340 (60.8%) | < 0.01 | 198 (56.3%) | 196 (55.7%) | 0.88 | |
| Cancer | 191 (21.3%) | 175 (29.8%) | < 0.01 | 117 (33.2%) | 93 (26.4%) | 0.05 | |
| Diabetes mellitus | 188 (21.0%) | 120 (22.3%) | 0.55 | 78 (22.2%) | 70 (20.0%%) | 0.48 | |
| Smoking | 134 (15.0%) | 118 (22.0%) | < 0.01 | 71 (20.2%) | 66 (18.8%) | 0.63 | |
| Coronary artery disease | 67 (7.5%) | 52 (9.8%) | 0.12 | 40 (11.4%) | 37 (10.5%) | 0.72 | |
| Chronic obstructive pulmonary disease | 54 (6.0%) | 87 (16.3%) | < 0.01 | 40 (11.4%) | 48 (13.6%) | 0.36 | |
| Non -dialysis chronic kidney disease | 48 (5.4) | 11 (2.1%) | < 0.01 | 8 (2.3%) | 8 (2.3%) | 1.00 | |
| Alcoholism | 46 (5.1%) | 39 (7.4%) | 0.08 | 16 (4.5%) | 20 (5.7%) | 0.50 | |
| Stroke | 27 (3.0%) | 37 (7.0%) | < 0.01 | 17 (4.8%) | 19 (5.4%) | 0.73 | |
| Anemia | 15 (1.7%) | 67 (11.4%) | < 0.01 | 12 (3.4%) | 15 (4.3%) | 0.55 | |
| Heart failure | 15 (1.7%) | 14 (2.7%) | 0.19 | 5 (1.4%) | 10 (2.9%) | 0.19 | |
| Other comorbidities | 251 (28.0%) | 47 (8.0%) | < 0.01 | 43 (12.2%) | 40 (11.4%) | 0.73 | |
| Number of comorbidities, median (IQR) | 2.0 (1.0–3.0) | 1.5 (1.0–3.0) | 0.02 | 2.0 (1.0–3.0) | 2.0 (1.0–3.0) | 0.23 | |

NOTES: SAPS 3 (Simplified Acute Physiology Score 3), SOFA (Sequential Organ Failure Assessment), ASA physical status classification: 2–4 (American Society of Anesthesiology), IQR (interquartile range).

The overall mean duration of surgery was 4.5 ± 2.5 hours. Head and neck surgeries, and orthopedic surgeries were more common in the 2018 cohort than in 2008. On the other hand, abdominal and vascular surgeries were more frequent in the 2008 cohort. After PSM, there were no difference in surgical procedures between cohorts.

In the full 2008 cohort, patients received more fluids and blood transfusions. Their serum creatinine and arterial lactate values were higher, while MAP values were lower as well as the use of vasopressors. Similar findings were observed after 1:1 PSM. Interestingly, the urine output at the end of surgery was higher in the 2008 study in all analyses, and the temperature was lower at the end of surgery (Table 2).

When comparing the stays in the full cohorts, the 2008 cohort had higher rates of complications and length of stay ICU and hospital. (Table 3).

**Table 2. Intraoperative data from the 2008 and 2018 studies in the full cohorts and after propensity score matching.**

| | Full cohort | | | Matched cohort | | |
|---|---|---|---|---|---|---|
| | 2018 (n = 904) | 2008 (n = 587) | P | 2018 (n = 352) | 2008 (n = 352) | P |
| **Type of surgery, n (%)** | | | 0.63 | | | 0.07 |
| Elective | 613 (69.2%) | 413 (70.4%) | | 255 (72.4%) | 276 (78.4%) | |
| Urgency | 273 (30.8%) | 174 (29.6%) | | 97 (27.6%) | 76 (21.6%) | |
| **Surgeries, n (%)** | | | | | | |
| Neurosurgery | 186 (20.8%) | 6 (1.0%) | < 0.01 | 0(0.0%) | 0 (0.0%) | ———— |
| Head and neck | 39 (4.4%) | 12 (2.0%) | 0.02 | 17 (4.8%) | 8 (2.3%) | 0.07 |
| Thoracic | 53 (5.9%) | 30 (5.1%) | 0.51 | 22 (6.3%) | 26 (7.4%) | 0.55 |
| Abdominal | 252 (28.1%) | 264 (45.0%) | < 0.01 | 138 (39.2%) | 138 (39.2%) | 1.00 |
| Vascular | 74 (8.3%) | 101 (17.2%) | < 0.01 | 41 (11.6%) | 49 (13.9%) | 0.37 |
| Orthopedic | 143 (16.0%) | 42 (7.2%) | < 0.01 | 31 (8.8%) | 31(8.8%) | 1.00 |
| Urologic | 48 (5.4%) | 34 (5.8%) | 0.72 | 28 (8.0%) | 21 (6.0%) | 0.30 |
| Others | 109 (11.1%) | 98 (16.7%) | < 0.01 | 31 (21.5%) | 76 (22.9%) | 0.65 |
| **Duration of surgery (min), median (IQR)** | 240.0 (180.0–360.0) | 230.0 (161.2–300.0) | < 0.01 | 240. (180.0–360.0) | 240.0 (180.0–315.0) | 0.33 |
| **Fluid administration** | | | | | | |
| Volume of crystalloids (ml), median (IQR) | 2400 (1500–3500) | 3000 (2000–5125) | < 0.01 | 2500 (1500–3500) | 3000 (2000–5000) | < 0.01 |
| Colloid volume (ml), median (IQR) | 500 (200–1000) | 500 (500–1000) | < 0.01 | 500 (200–1000) | 500 (500–1000) | 0.13 |
| Crystalloids: Colloid volume (ml), median (IQR) | 9.5 (3.9–19.25) | 7.0 (4.0–12.0) | 0.24 | 9.0 (3.2–13.0) | 7.0 (3.0–12.0) | 0.06 |
| Total volume (ml), median (IQR) | 2500 (1500–3500) | 3500 (2000–5500) | < 0.01 | 2500 (1500–3500) | 3000 (2000–5500) | < 0.01 |
| Units of packed red blood cells (IU), median (IQR) | 2 (1.0–2.25) | 2.0 (2.0–3.0) | 0.02 | 2.0 (1.0–3.0) | 2.0 (1.0–3.0) | 0.19 |
| Urine output (ml), median (IQR) | 1000 (600–1500) | 1635 (1100–2300) | < 0.01 | 900 (500–1400) | 1600 (1100–2211) | < 0.01 |
| Water balance (ml), median (IQR) | 1900 (1100–3000) | 1750 (400–3946) | 0.13 | 1650 (1100–3200) | 1975 (400–3901) | 0.04 |
| **Serum creatinine at the end of surgery (mg/dl), median (IQR)** | 0.8 (0.6–1.1) | 1.0 (0.8–1.4) | < 0.01 | 0.8 (0.6–1.2) | 0.9 (0.8–1.2) | 0.04 |
| **Need for blood transfusion, n (%)** | 141 (16.1%) | 174 (29.6%) | < 0.01 | 80 (22.7%) | 100 (28.4%) | 0.04 |
| **Axillary temperature at the end of surgery (˚C), mean±SD** | 35.8±0.9 | 35.5±0.8 | < 0.01 | 35.7±0.9 | 35.5±0.9 | 0.04 |
| **Need for vasopressors, n (%)** | 157 (18.0%) | 94 (16.1%) | 0.35 | 82 (23.3%) | 45 (12.8%) | < 0.01 |
| **MAP (mmHg), mean±SD during the intraoperative period** | 90.5±18.3 | 72.6±15.5 | < 0.01 | 89.6±19.4 | 72.3±15.1 | < 0.01 |
| **Arterial lactate at the end of surgery (mmol/L), median (IQR)** | 1.8 (1.2–2.7) | 2.2 (1.3–3.6) | < 0.01 | 1.8 (1.2–2.8) | 2.1 (1.0–3.4) | 0.03 |

NOTES: Other surgeries (ophthalmology, plastics, gynecology and transplants), IQR (interquartile range), ml (milliliter), mmol (millimoles), L (liter), mg (milligram), dl (deciliter), n (number of events), ˚C degrees Celsius, mmHg (millimeters of mercury), SD (standard deviation).

**Table 3. Comparison of complications and length of stay in the postoperative period.**

| | Full cohort | | | Matched Cohort | | |
|---|---|---|---|---|---|---|
| | **2018 (n = 904)** | **2008 (n = 587)** | **P** | **2018 (n = 352)** | **2008 (n = 352)** | **P** |
| **Total complications, n (%)** | 265 (29.9%) | 225 (38.3%) | < 0.01 | 74 (21.0%) | 146 (41.5%) | < 0.01 |
| **Length of ICU stay (days), median (IQR)** | 2.0 (1.0–4.0) | 2.0 (1.0–5.0) | < 0.01 | 2.0 (1.0–4.0) | 2.1 (1.1–4.0) | < 0.01 |
| **Length of hospital stay (days), median (IQR)** | 9.5 (5.4–18.6) | 10.0 (4.0–21.0) | 0.86 | 9.6 (5.6–19.2) | 10 (4.0–20.0) | 0.60 |

NOTES: IQR (interquartile range), ICU (intensive care unit), n (number of events).

After the PSM, infectious was the most different outcome between the cohorts, even taking into account adjustments. Patients from the 2008 period had more postoperative complications when we analyzed the total effect of negative outcomes (OR 1.6, 95% CI 1.4–1.9) (Fig 3).

There was no difference in the hospital length of stay between the 2018 and 2008 cohorts: 9.6 days (5.6–19.2) vs. 10 days (4–20); P = 0.60, was not significant. After PSM, the ICU length of stay was significantly shorter in the 2018 cohort, as was mortality in the ICU and hospital (Table 3). Thus, the risk ratio for mortality in 2008, with adjustments for previous disease (cancer) was as follows: in the ICU, hazard ratio (HR) = 2.39 (95% CI: 1.36–4.20), and in the hospital, HR = 1.64 (95% CI: 1.03–2.62) for the patients in the 2008 cohort (Fig 4).

Performing a logistic regression analysis with the full cohort adjusted by the propensity score of the significant variables between the two periods, we noted that the risk of death brought by every 100-ml increase in fluid balance was OR = 1.03 (95% CI 1.01–1.06), and for each 1-point increase in creatinine, the OR was 1.31 (95% CI 1.1–1.56). For patients who

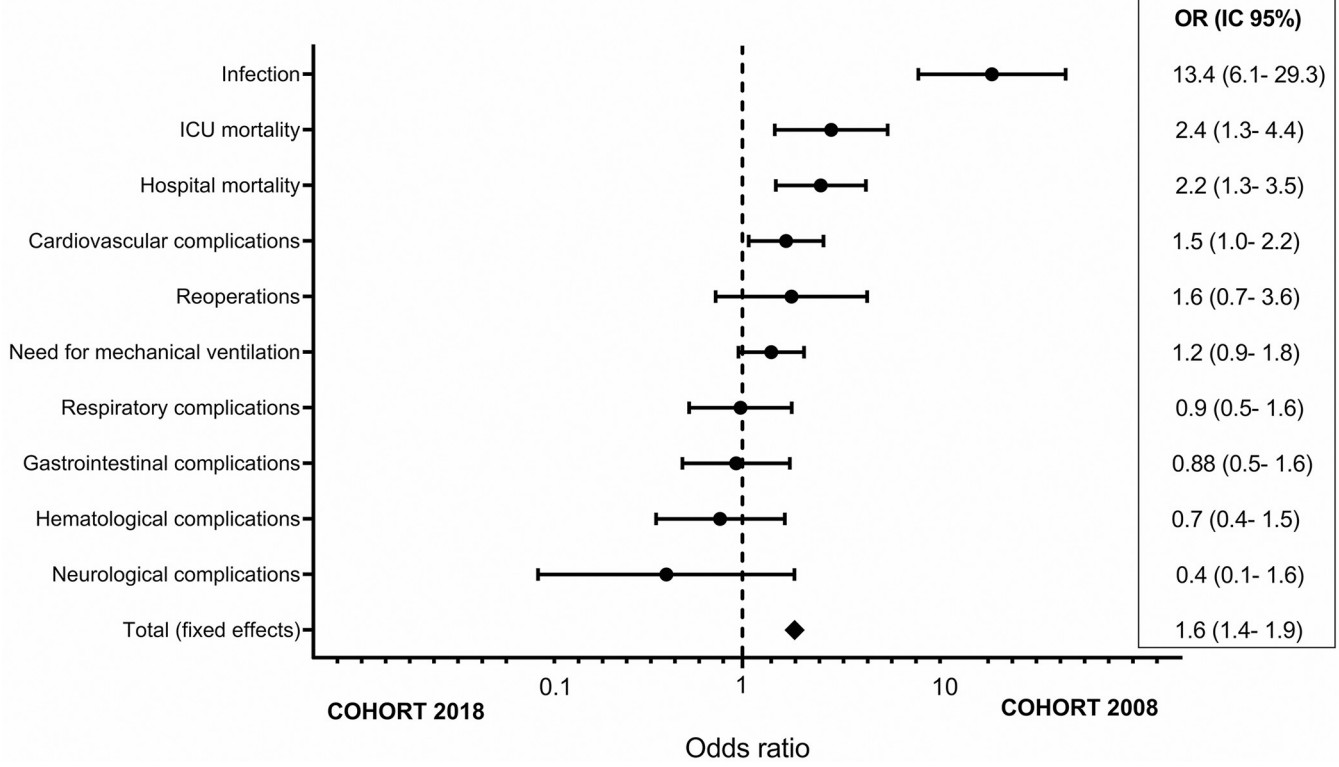

**Fig 3. Effect of negative outcomes in the 2018 and 2008 cohorts, with adjustments of the confounding factors among the baseline characteristics.**

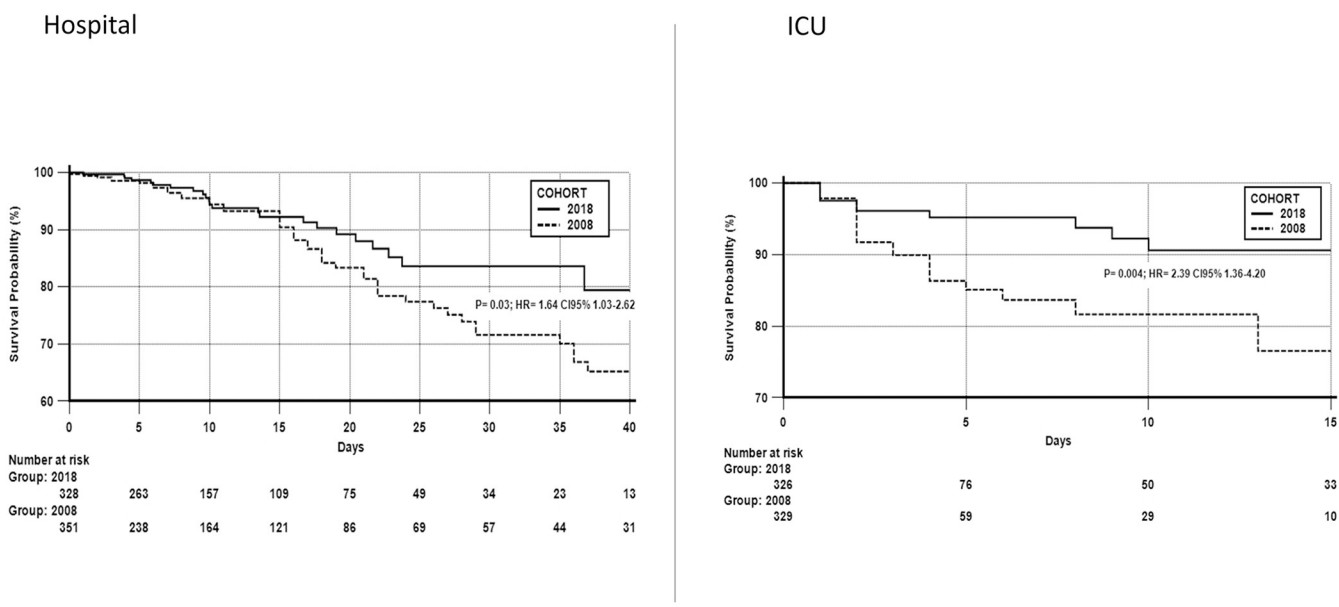

**Fig 4. ICU and hospital survival curves comparing the 2018 and 2008 cohorts after PSM.**

received blood transfusion during surgery, the OR was 2.23 (95% CI 1.35–4.0). On the other hand, for each 1-mmHg increase in MAP, according to the limits of sample values from 43 mmHg to 118 mmHg, the risk decreased (OR = 0.96, 95% CI 0.95–0.98). It was not observe an association between hospital mortality and other variables (Table 4).

## Discussion

In this post-hoc analysis of two Brazilian multicenter prospective cohort studies, a large group of surgical patients admitted to the ICU in the postoperative period was analyzed. The results

**Table 4. Factors correlated with hospital mortality from among the significant variables in the adjusted comparison of the cohorts (univariate and multivariate analysis).**

|  | Univariate | | | Multivariate | | |
|---|---|---|---|---|---|---|
|  | OR | 95% CI | P | OR | 95% CI | P |
| Chronic diseases: |  |  |  |  |  |  |
| Cancer | 0.690 | 0.406–1.172 | 0.169 |  |  |  |
| Temperature at the end of surgery (˚C) | 0.734 | 0.571–0.942 | 0.015 | 0.838 | 0.639–1.100 | 0.204 |
| Intraoperative volume of crystalloids (ml) | 1.0001 | 1.000–1.0001 | 0.055 |  |  |  |
| Intraoperative fluid balance (per 100 ml) | 1.011 | 1.003–1.019 | 0.007 | 1.03 | 1.01–1.06 | 0.003 |
| MAP (mmHg) | 0.966 | 0.953–0.981 | < 0.001* | 0.968 | 0.952–0.984 | < 0.001 |
| Need for vasopressors | 5.853 | 3.600–9.515 | < 0.001* |  |  |  |
| Need for packed red blood cells | 3.955 | 2.483–6.298 | < 0.001 | 2.232 | 1.350–4.000 | 0.002 |
| Creatinine (per mg/dl) | 1.504 | 1.282–1.764 | < 0.001* | 1.314 | 1.105–1.563 | 0.002 |
| Urine output (per ml) | 0.999 | 0.9994–1.000 | 0.053* |  |  |  |

NOTES: ml (milliliter), mmol (millimoles), L (liter), mg (milligram), dl (deciliter), n (number of events), ˚C (degrees Celsius), mmHg (millimeters of mercury), OR (risk ratio), 95% CI (95% confidence interval), OR (risk ratio), P (p value).

*Collinearity between variables.

indicate that an improvement in intensive care practices in surgical patients have occurred during a 10-year interval.

In the recent cohort of 2018, most favorable outcomes were due to a better fluid and transfusion management, suitable hemodynamic protocols including higher blood pressure levels, earlier use of vasoactive drugs and hence lower lactate levels. Contemporary literature promotes the use of rational and protocol hemodynamic monitoring in the perioperative period, with the aim of reducing the length of stay in the ICU and hospital, reducing the duration of mechanical ventilation and reducing complications [8]. The FEDORA study demonstrated that goal-directed hemodynamic management in surgical patients led to fewer infections, fewer non-cardiogenic pulmonary edema, and a lower incidence of acute kidney injury [8,9].

The patients in the 2008 cohort had more postoperative complications. When the cohorts were balanced, the 2008 cohort had higher infection and cardiovascular complications rates. Furthermore, ICU stay, ICU and hospital mortality were higher, and the overall negative outcome effect was OR = 1.6 (95% CI 1.4–1.9).

When data from both cohorts were analyzed together, some variables were strongly associated with hospital mortality, such as intraoperative blood transfusion, intraoperative MAP, serum creatinine at the end of surgery, and excessive intraoperative fluid balance. However, these factors were different between the balanced 2018 and 2008 cohorts, maybe justifying the outcomes difference.

Greater packed red blood cell transfusion in the intraoperative period was associated with higher mortality, as already seen priorly [10–13]. This difference could be due to the lower transfusion trigger in 2018.

Besides, lower MAP in the immediate postoperative period was correlated with higher mortality. A randomized controlled study from 2017 identified that a more individualized management of MAP in the intraoperative period generated fewer organ dysfunctions in the postoperative period [14]. These findings strengthen the actions directed at early hemodynamic management and individualization of clinical measures [14,15]. In this present analysis, we noted that patients in the 2018 cohort received more vasoactive drugs in the perioperative period. This finding might be due to the understanding that the early use of these drugs can more quickly increase the MAP and improve the perfusion [16–18]. Vasoactive drugs used to adjust MAP have not been associated with complications [19].

Excessive intraoperative fluid balance was correlated with a higher mortality. Similar findings were observed in several other studies that have also associated higher fluid balance with worse outcomes and have carefully considered fluid management in surgical patients [20–22]. Something that caught our attention was that the same volume of synthetic colloid solutions was administered in both cohorts. This clearly demonstrates that the prescriptive practice of colloids in surgical patients has not changed over the years, despite evidence of their negative effects in sepsis patients [23,24].

Nonetheless, high-risk surgery patients are the victims of most perioperative morbidity and mortality [25] and among the reasons for their unfortunate outcomes is inadequate hemodynamic optimization [26], which involves adjustments in blood pressure, cardiac output, blood volume, and cardiac contractility [27]. These adjustments require specialized handling and an adequate level of monitoring, with well-defined protocols and well-defined goals [28]. Fluid administration is generally accepted as a standard for the resuscitation of critically ill patients. However, since excessive fluid administration is associated with higher mortality [29], an individualized assessment of fluid responsiveness is advocated as good practice [30].

Monitoring and maximizing the systolic volume per fluid load during high-risk surgery are associated with a better postoperative outcome [31,32]. This can alert clinicians to blood pressure variations, which aids in proper volume adjustment [33].

As for the metabolic variables, higher creatinine in the immediate postoperative period was associated with mortality. Acute kidney injury (AKI) is a condition that reflects the context of patient severity within the ICU and is associated with worse outcomes and mortality [34]. It is no different in surgical patients, as has been the subject of recent studies that examined the correlation of poor prognosis with higher levels of acute kidney injury [35–37]. Although patients in the 2008 cohort had higher urine volume, the diagnosis of acute kidney injury is always determined by the value of creatinine and/or urine output [38].

In addition, hypothermia was statistically different between the cohorts in the univariate analysis, and may have contributed to the poor results of the 2008 patients, like discussed in literature [39].

One subject that we need to address that may have directly improved the outcomes of the 2018 cohort is advances in multidisciplinary care within Brazilian ICUs [40]. There have also been significant advances in our knowledge about perioperative care, with practices becoming increasingly personalized and supported by evidence-based medicine [3]. Protocols and checklists used in surgical patients have greatly improved in recent years while the overall quality of the centers that serve this patient population has also improved [41].

## Study limitations and strengths

This study has some shortcomings that should be contextualized. There is no specific protocol that encompasses surgeries performed by the centers involved in the study. There were also no homogenized hemodynamic and transfusion optimization protocols established among all centers. Furthermore, for the fluid balance calculation, it was not considered insensitive losses due to the difficulty in estimating them and the definition of infection was based on the record of the diagnosis of infection in addition to the use of antibiotics and positive cultures, a fact that despite not being a unanimous definition, it was able to homogenize the two study cohorts. In addition, we failed to capture some relevant data that could be included in the analyses, such as postoperative data on ICU care, but this was not the main objective of this study because surgical patients had short ICU stays.

Within the methodological content of our study, we know that the statistical method of PSM has deficiencies, as it can cause patient losses, and inaccurate results, but every care was taken to avoid these possible failures [42]. A rigorous model of PSM were considered in our study to reduce the heterogeneity between the groups and thus generate accurate and trustworthy results. However, the fact that we had minimal differences in the variables between the groups and the support of the main outcomes of the full cohorts after pairing showed that the cohorts were well balanced and therefore our statistical methods are likely to provide reliable results.

Our study had some strengths that we should consider, such as the selection of a large random sample of Brazilian ICUs from all regions of the country. This fact made our sample representative. However, our study model is not robust enough as to be conclusive on the data at the national level, but this model can be used as a comparison for potent future national epidemiological studies with similar objectives in Brazil or other countries, with even larger databases. Finally, this study updates the improvements of care over the years in high-risk surgical patients in Brazil.

## Conclusion

From 2008 to 2018, the mortality of high-risk surgical patients in Brazilian ICUs decreased. A lower MAP and higher creatinine in the immediate postoperative period, greater fluid balance and greater need for red blood cell transfusions in the intraoperative period were the

significant differences found in the mortality trends between decades. Cardiovascular complications and infections showed a decrease in their incidence.

The most recent findings indicate that care has changed over time, a fact that may have directly impacted the outcomes of surgical patients.

## Supporting information

**S1 Data.**
(XLS)

## Acknowledgments

The following have made substantial contributions to the study. We would like to thank the important work and support of the BraSIS research group *(brasis@googlegroups.com)*, which carried out data collection and care for patients and their data for research: Murillo Santucci Cesar Assunção (Hospital Israelita Albert Einstein, SP, Brazil); Ary Serpa Neto (Hospital Israelita Albert Einstein, SP, Brazil); Neymar Elias de Oliveira (Hospital de Base—São José do Rio Preto, SP, Brazil); Viviane Cordeiro Veiga (Beneficiência Portuguesa, SP, Brazil); Salomón Soriano Ordinola Rojas (Beneficiência Portuguesa, SP, Brazil); Fabio Eduardo Bosso (Universidade Cidade de São Paulo, SP, Brazil); José Mário Teles (Hospital de Urgências de Goiânia, GO, Brazil); Alexandre Amaral (Hospital de Urgências de Goiânia, GO, Brazil); Bruno Melo Nobrega de Lucena (Hospital das Clinicas de São Paulo, SP, Brazil); Raphael Augusto Gomes de Oliveira (Hospital das Clinicas de São Paulo, SP, Brazil); Luciana Coelho Sanches (Hospital do Câncer de Barretos, SP, Brazil); Antônio Paulo Nassar Junior (Hospital AC Camargo, SP, Brazil); Álvaro Réa-Neto (CEPETI, PR, Brazil); Flávio Geraldo Rezende de Freitas (Hospital SEPACO, SP, Brazil); Antônio Tonete Bafi (Hospital SEPACO, SP, Brazil); Eduardo Souza Pacheco (Hospital SEPACO, SP, Brazil); Fernando José Ramos (Hospital Sírio Libanês, SP, Brazil); Maria Augusta Santos Rahe Pereira (Hospital Santa Casa de Campo Grande, MS, Brazil); Danielle Dourado Magalhães (Hospital Beneficiente São Vicente de Paulo, RS, Brazil); Diogo Oliveira Toledo (Hospital e Maternidade São Luiz do Itaim, SP, Brazil); Fernando Suparregui Dias (Hospital Pompeia, RS, Brazil); Natalia Fioravante Postalli (Hospital Beneficência Portuguesa, SP, Brazil); Ricardo Azevedo Cruz D'Oliveira (Hospital Português da Bahia, BA, Brazil); André Ricardo de Oliveira Estrela (Hospital Português da Bahia, BA, Brazil); Fábio Sartori Schwerz (Associação Beneficente de Campo Grande, MS, Brazil); Giovanna Padoa de Menezes (Associação Beneficente de Campo Grande, MS, Brazil); Ulysses Vasconcellos de Andrade e Silva (Hospital de Câncer de Barretos, SP, Brazil); Thais Kawagoe Alvarisa (Hospital Beneficência Portuguesa, SP, Brazil); André Meregalli (Irmandade da Santa Casa de Misericórdia de Porto Alegre, RS, Brazil); Jorge Amilton Höher (Irmandade da Santa Casa de Misericórdia de Porto Alegre, RS, Brazil); Marciano de Sousa Nobrega (Hospital das Clínicas, Universidade Federal de Goiás, GO, Brazil); Claudio Piras (Vitoria Apart Hospital, ES, Brazil); Stéphanie de Barros Piras (Vitoria Apart Hospital, ES, Brazil); Rodrigo Conti (Vitoria Apart Hospital, ES, Brazil); Paulo Lisboa Bittencourt (Hospital Português da Bahia, BA, Brazil); José Mauro Vieira Júnior (Hospital Sirio Libanês, SP, Brazil); Mirella Cristine de Oliveira (CEPETI, PR, Brazil); Luana Tannous (CEPETI, PR, Brazil); Rafael Alexandre de Oliveira Deucher (CEPETI, PR, Brazil); Henrique Tadashi Katayama (Hospital do Servidor Público Estadual, SP, Brazil); Marcos Henrique Borges Ferreira (Hospital Nossa Senhora de Fátima, MG, Brazil); Wagner Luis Nedel (Hospital Nossa Senhora da Conceição, RS, Brazil); Matheus Golenia dos Passos (Hospital Nossa Senhora da Conceição, RS, Brazil); Luiz Gustavo Marin Hospital (Nossa Senhora da Conceição, RS, Brazil); Cristine Pilati Pileggi Castro (Hospital São Vicente

de Paulo, RS, Brazil); Sabrina Frighetto Henrich (Hospital São Vicente de Paulo, RS, Brazil); Bruna Fernanda Camargo Silva Parra (Hospital e Maternidade São Luiz Itaim, SP, Brazil); Luiza Zerman (Hospital Pompeia, RS, Brazil); Fernanda Formolo (Hospital Pompeia—Caxias do Sul, RS, Brazil); Denner Luiz Vilela (Hospital Nossa Senhora de Fátima, MG, Brazil); Guilherme Cincinato de Almeida (Hospital Nossa Senhora de Fátima, MG, Brazil); Wilson de Oliveira Filho (Hospital e Pronto Socorro 28 de Agosto, AM, Brazil); Raoni Machado Coutinho (Hospital e Pronto Socorro 28 de Agosto, AM, Brazil); Michele Cristina Lima de Oliveira (Hospital e Pronto Socorro 28 de Agosto, AM, Brazil); Gilberto Friedman (Irmandade da Santa Casa de Misericórdia de Porto Alegre, RS, Brazil); and Afonso José Celente Soares (Hospital de Força Aérea do Galeão, RJ, Brazil).

The present study was endorsed by the Brazilian Research in Intensive Care Network (AMIBnet).

## Author Contributions

**Conceptualization:** Brenno Cardoso Gomes, João Manoel Silva Júnior.

**Data curation:** Brenno Cardoso Gomes, Suzana Margareth Ajeje Lobo, Luiz Marcelo Sá Malbouisson, Renato Carneiro de Freitas Chaves, Thiago Domingos Corrêa, João Manoel Silva Júnior.

**Formal analysis:** Brenno Cardoso Gomes, Luiz Marcelo Sá Malbouisson, João Manoel Silva Júnior.

**Investigation:** Brenno Cardoso Gomes, Suzana Margareth Ajeje Lobo, Luiz Marcelo Sá Malbouisson, Renato Carneiro de Freitas Chaves, Thiago Domingos Corrêa, Cristina Prata Amendola, João Manoel Silva Júnior.

**Methodology:** Brenno Cardoso Gomes, João Manoel Silva Júnior.

**Project administration:** Brenno Cardoso Gomes, João Manoel Silva Júnior.

**Resources:** Brenno Cardoso Gomes, João Manoel Silva Júnior.

**Software:** Brenno Cardoso Gomes.

**Supervision:** Brenno Cardoso Gomes, Luiz Marcelo Sá Malbouisson, João Manoel Silva Júnior.

**Validation:** Brenno Cardoso Gomes, Luiz Marcelo Sá Malbouisson, João Manoel Silva Júnior.

**Visualization:** Brenno Cardoso Gomes.

**Writing – original draft:** Brenno Cardoso Gomes.

**Writing – review & editing:** Brenno Cardoso Gomes, Suzana Margareth Ajeje Lobo, Luiz Marcelo Sá Malbouisson, João Manoel Silva Júnior.

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
