## [Decision Letter · Decision Letter 0]

1 Mar 2023

PONE-D-23-01398Trends in perioperative practices of high-risk surgical patients over a 10-year interval.PLOS ONE

Dear Dr. Gomes,

Thank you for submitting your manuscript to PLOS ONE. After careful consideration, we feel that it has merit but does not fully meet PLOS ONE’s publication criteria as it currently stands. Therefore, we invite you to submit a revised version of the manuscript that addresses the points raised during the review process.

ACADEMIC EDITOR:  Thank you for the article. Let me give you a comment. The fact that in 10 years medicine and surgery have improved in terms of prevention, surgical treatments, and therapies is undeniable. This is neither new nor exciting news. Certainly the research is well conducted, the statistical analysis correct, and the result good. Nevertheless it is necessary to correct some parts as requested by the reviewers, expecially reviewer 2No conflicts between the reviewsI recommend also a bibliography check. Please submit your revised manuscript by Apr 15 2023 11:59PM. If you will need more time than this to complete your revisions, please reply to this message or contact the journal office at plosone@plos.org. Please include the following items when submitting your revised manuscript:A rebuttal letter that responds to each point raised by the academic editor and reviewer(s). You should upload this letter as a separate file labeled 'Response to Reviewers'.A marked-up copy of your manuscript that highlights changes made to the original version. You should upload this as a separate file labeled 'Revised Manuscript with Track Changes'.An unmarked version of your revised paper without tracked changes. You should upload this as a separate file labeled 'Manuscript'.

We look forward to receiving your revised manuscript.

Kind regards,

Martina Crivellari

Academic Editor

PLOS ONE

Journal Requirements:

3. One of the noted authors is a group or consortium BraSIS research group. In addition to naming the author group, please list the individual authors and affiliations within this group in the acknowledgments section of your manuscript. Please also indicate clearly a lead author for this group along with a contact email address.

4. We note that Figure 2 in your submission contain map image which may be copyrighted. All PLOS content is published under the Creative Commons Attribution License (CC BY 4.0), which means that the manuscript, images, and Supporting Information files will be freely available online, and any third party is permitted to access, download, copy, distribute, and use these materials in any way, even commercially, with proper attribution. For these reasons, we cannot publish previously copyrighted maps or satellite images created using proprietary data, such as Google software (Google Maps, Street View, and Earth). For more information, see our copyright guidelines: http://journals.plos.org/plosone/s/licenses-and-copyright.

Reviewers' comments:

Reviewer's Responses to Questions

**Comments to the Author**

1. Is the manuscript technically sound, and do the data support the conclusions?

Reviewer #1: Yes

Reviewer #2: Yes

2. Has the statistical analysis been performed appropriately and rigorously? 

Reviewer #1: Yes

Reviewer #2: Yes

3. Have the authors made all data underlying the findings in their manuscript fully available?

Reviewer #1: Yes

Reviewer #2: Yes

4. Is the manuscript presented in an intelligible fashion and written in standard English?

Reviewer #1: Yes

Reviewer #2: Yes

5. Review Comments to the Author

Reviewer #1: Paper well written, with consistent results and conclusions.

Just 2-3 minor revisions:

1. in the fluid balance was drain losses and perspiratio calculated?Please specify and discuss, in case, as limit

2. Just revise some adverbes (such as in the perioperatively, prefer in the perioperative period)

3. Maybe It would help to add a sentence in the discussion section about the net effect of the interest in the hemodynamics in the preoperative period in these 10 years. The concept of monitoring hemodynamic de-arrangements and target the intervention to hemodynamic goals, to personalise fluid and vasopressor therapy with appropriate monitoring, basing on surgical and patient risk, in order to avoid both hypo and hyper fluid resuscitation, and to preserve organ perfusion is strongly emerged in this decade. I think that this is one of the most important component that explain your results, both in terms of MAP and fluid overload. Moreover, also greater urine output in the 2008 period could be justified by the greater amount of fluids given (see quantile ranges), but this this greater amount of fluids has no impact on renal function , as creatinine values between the 2 groups show. This observation further confirms that fluids and MAP should not be considered as separate entities in the preoperative period, but that the best way to protect organ function is a protocolized hemodynamic strategy that considers fluids, MAP and CO as linked variables that all contribute to influence organ perfusion.

Reviewer #2: Dear authors, congratulation for your work.

I didn't find any ethics concerns about your analysis of surgical ICU admitted patients in two different periods.

I have some concerns and suggestions:

In abstract (line 38…) it is not clear if correlation between MAP and survival rate is linear or if an upper limit exists.

Line 40 is unclear, please consider modifying it.

Line 130: can you please explain which definition of infection did you use? I think it is important for us to know it since it is a major difference point between the cohorts (lines 295-299).

Line 263, it is unclear. You found 9.6 days of in-hospital length of stay in 2008 cohort and 10 days in 2018 cohort. Why did you say, in line 262, that in-hospital length of stay was difference? Please consider modifying it.

Regarding factor linked to in hospital mortality, it is not clear how you managed red blood cells transfusions. As you know, both excessive and insufficient transfusions are known death risk factors. Which were the threshold limits to start a red blood cells transfusion? Which was the target to stop transfusions? If you have this information, please add it to manuscript or specify you don’t have it. In line 293 you say that there was a better transfusion management in 2018 cohort, can you explain it, please?

Since you describe the impact of vasoactive drug on survival rates may you add some information about the vasoactive drugs management (thresholds, MAP targets, drugs used…)?

Line 327 It would be wonderful if you add information about crystalloids : colloids volumes ratio.

All the best.

6. PLOS authors have the option to publish the peer review history of their article (what does this mean?). If published, this will include your full peer review and any attached files.

Reviewer #1: **Yes: **Mariateresa giglio

Reviewer #2: **Yes: **Gaetano Lombardi

---

## [Author Response · Author response to Decision Letter 0]

23 Mar 2023

Data, 03/09/2023

Dear editor,

Martina Crivellari.

Academic Editor.

PLOS ONE.

We would like to thank you for the recommendations given for the article. All were of great value to improve our study. We also appreciate the opportunity and the possibility of having our article published in this journal. Below is the summary of the questions and opinions asked by the reviewers and our answers.

Response to Reviewers.

Review Comments to the Author.

Reviewer #1: Paper well written, with consistent results and conclusions.

Just 2-3 minor revisions:

I greatly appreciate the considerations and we add all the suggestions.

1. in the fluid balance was drain losses and perspiration calculated? Please specify and discuss, in case, as limit.

R: Thank you very much for the question. The insensitive losses were not calculated, as the calculation is not homogeneously unanimous as to how it could be exactly estimated. However, we will consider the limitations of the study. On the other hand, the loss of drains was considered.

2. Just revise some adverbes (such as in the perioperatively, prefer in the perioperative period).

R: Yes, thank you. We made the adjustment.

3. Maybe It would help to add a sentence in the discussion section about the net effect of the interest in the hemodynamics in the preoperative period in these 10 years. The concept of monitoring hemodynamic de-arrangements and target the intervention to hemodynamic goals, to personalize fluid and vasopressor therapy with appropriate monitoring, basing on surgical and patient risk, in order to avoid both hypo and hyper fluid resuscitation, and to preserve organ perfusion is strongly emerged in this decade. I think that this is one of the most important component that explain your results, both in terms of MAP and fluid overload. Moreover, also greater urine output in the 2008 period could be justified by the greater amount of fluids given (see quantile ranges), but this this greater amount of fluids has no impact on renal function, as creatinine values between the 2 groups show. This observation further confirms that fluids and MAP should not be considered as separate entities in the preoperative period, but that the best way to protect organ function is a protocolized hemodynamic strategy that considers fluids, MAP and CO as linked variables that all contribute to influence organ perfusion.

R: Excellent aspect put by the reviewer. Yes, we totally agree that the care of these serious patients must be protocoled and supported by refined and individualized hemodynamic goals. We also agree that uncontrolled fluid balance in critically ill patients causes an increase in complications and a worse prognosis, a condition that also occurs in high-risk surgical patients. In the renal aspect, there are several studies that associate a positive fluid balance with no improvement in renal function and even with deterioration of renal function, leading to increased venous pressures and renal congestion. Really better hemodynamic management of this group of patients has been gaining strength in recent years with rich literature (Katayama HT, Gomes BC, Lobo SMA, Chaves RCF, Corrêa TD, Assunção MSC, Serpa Neto A, Malbouisson LMS, Silva-Jr JM; BraSIS study group Investigators. The effects of acute kidney injury in a multicenter cohort of high-risk surgical patients. Ren Fail. 2021 Dec;43(1):1338-1348. doi: 10.1080/0886022X.2021.1977318. PMID: 34579622; PMCID: PMC8477947).

We have added a thread on the subject to the discussion. Thank you very much!

Reviewer #2: Dear authors, congratulation for your work.

I didn't find any ethics concerns about your analysis of surgical ICU admitted patients in two different periods. I have some concerns and suggestions:

Thank you very much for the questions that certainly added a lot to this study.

1. In abstract (line 38…) it is not clear if correlation between MAP and survival rate is linear or if an upper limit exists.

R: By increasing the mean arterial pressure, according to the limits of sample values from 43 mmHg to 118 mmHg, the risk of death decreased.

In this probability graph, it can be seen that as the MAP values increase, the probability of death decreases, however the probability was checked within the inferiority and superiority limits 43 to 118 mmHg. That was added in the abstract.

2. Line 40 is unclear, please consider modifying it.

R: Sorry for our mistake. We fixed the text. Thank you very much.

3. Line 130: can you please explain which definition of infection did you use? I think it is important for us to know it since it is a major difference point between the cohorts (lines 295-299).

R: Important aspect you mentioned. Thanks. As we analyzed formatted databases, historically (from 2008 and 2018), we chose to consider infection in the groups, when the presence of “infection” was documented in the spreadsheets used in the research, together with the use of antibiotics and the presence of any positive cultures. Regardless of whether it is infection, sepsis or septic shock. In this way, we believe to homogenize the diagnosis.

We have briefly commented on this on lines 134-136. But added this as a potential limitation.

4. Line 263, it is unclear. You found 9.6 days of in-hospital length of stay in 2008 cohort and 10 days in 2018 cohort. Why did you say, in line 262, that in-hospital length of stay was difference? Please consider modifying it.

R: We've fixed the phrase in the revised text for better understanding. Thanks.

5. Regarding factor linked to in hospital mortality, it is not clear how you managed red blood cells transfusions. As you know, both excessive and insufficient transfusions are known death risk factors. Which were the threshold limits to start a red blood cells transfusion? Which was the target to stop transfusions? If you have this information, please add it to manuscript or specify you don’t have it. In line 293 you say that there was a better transfusion management in 2018 cohort, can you explain it, please?

R: Thanks for the questions. There were no homogenized protocols for hemodynamic handling, general care and transfusion for patients. Thus, in none of the cohorts, we did not have transfusion targets and cutoff points for red blood cell prescription. On lines 361-363 we comment on this. But to make it clearer, we added in the article, specifically the information about the lack of unique transfusion protocols between the participating centers. In limitation of the study.

Being an observational study that analyzed, among other conditions, care practices in two large multicenter studies, we identified in the multivariate analysis that there was a correlation between transfusion and death. As the groups were paired and balanced, we believe that the fact that we had less transfusion in 2018 and less mortality, the action of prescribing fewer red blood cells was a protective practice. Therefore, we chose to write the comment on line 293.

The idea of our work was to have a general mapping in the two cohorts of the factors that were associated with a worse prognosis and that could have improved over time.

6. Since you describe the impact of vasoactive drug on survival rates may you add some information about the vasoactive drugs management (thresholds, MAP targets, drugs used…)?

R: Thanks for the note. As there was no single, predetermined protocol for hemodynamic monitoring in the centers involved, each location carried out its own intervention plan. Thus, we do not have concrete data on the most used drugs and their doses. Just like the previous question we added limitations.

Line 327 It would be wonderful if you add information about crystalloids: colloids volumes ratio.

R: Crystalloids: Colloid volume (ml), median (IQR) 

Full cohort 

2018: 9.5 (3.9- 19.25) 2008: 7.0 (4.0- 12.0) P=0.24 

Matched Cohort

2018:9.0 (4.0-12.0) 2008:7.0 (3.0- 12.0) P=0.51

It was included in table 2.

Brenno Cardoso Gomes.

Professor at the Department of Integrated Medicine at UFPR, Curitiba-PR, Brazil.

---

## [Decision Letter · Decision Letter 1]

16 May 2023

Trends in perioperative practices of high-risk surgical patients over a 10-year interval.

PONE-D-23-01398R1

Dear Dr. Gomes,

We’re pleased to inform you that your manuscript has been judged scientifically suitable for publication and will be formally accepted for publication once it meets all outstanding technical requirements.

Kind regards,

Samuele Ceruti

Academic Editor

PLOS ONE

Additional Editor Comments (optional):

Reviewers' comments:

Reviewer's Responses to Questions

**Comments to the Author**

1. If the authors have adequately addressed your comments raised in a previous round of review and you feel that this manuscript is now acceptable for publication, you may indicate that here to bypass the “Comments to the Author” section, enter your conflict of interest statement in the “Confidential to Editor” section, and submit your "Accept" recommendation.

Reviewer #1: All comments have been addressed

Reviewer #2: All comments have been addressed

2. Is the manuscript technically sound, and do the data support the conclusions?

Reviewer #1: Yes

Reviewer #2: Yes

3. Has the statistical analysis been performed appropriately and rigorously? 

Reviewer #1: Yes

Reviewer #2: Yes

4. Have the authors made all data underlying the findings in their manuscript fully available?

Reviewer #1: Yes

Reviewer #2: Yes

5. Is the manuscript presented in an intelligible fashion and written in standard English?

Reviewer #1: Yes

Reviewer #2: Yes

6. Review Comments to the Author

Reviewer #1: The paper is improved, the authors have answered to all questions. The method are correctly presented.

Reviewer #2: Dear authors congratulations for your work, I think it is a very interesting point of view on evolving care in critical care setting. Your answers solved all my concerns.

Thank you very much.

7. PLOS authors have the option to publish the peer review history of their article (what does this mean?). If published, this will include your full peer review and any attached files.

Reviewer #1: No

Reviewer #2: **Yes: **Gaetano Lombardi

---

## [Editor Report · Acceptance letter]

18 May 2023

PONE-D-23-01398R1 

Trends in perioperative practices of high-risk surgical patients over a 10-year interval 

Dear Dr. Gomes:

I'm pleased to inform you that your manuscript has been deemed suitable for publication in PLOS ONE. Congratulations! Your manuscript is now with our production department. 

Kind regards, 

on behalf of

Dr. Samuele Ceruti 

Academic Editor

PLOS ONE